# Differential Modulation of the Phosphoproteome by the MAP Kinases Isoforms p38α and p38β

**DOI:** 10.3390/ijms241512442

**Published:** 2023-08-04

**Authors:** Dganit Melamed Kadosh, Jonah Beenstock, David Engelberg, Arie Admon

**Affiliations:** 1Faculty of Biology, Technion—Israel Institute of Technology, Haifa 3200003, Israel; sdganit@technion.ac.il; 2Department of Biological Chemistry, Alexander Silberman Institute of Life Sciences, The Hebrew University of Jerusalem, Jerusalem 9190401, Israel; beenstock@lunenfeld.ca; 3Singapore-HUJ Alliance for Research and Enterprise, Mechanisms of Liver Inflammatory Diseases Program, National University of Singapore, Singapore 138602, Singapore; 4Department of Microbiology and Immunology, Yong Loo Lin School of Medicine, National University of Singapore, Singapore 117456, Singapore

**Keywords:** p38 MAPK, phosphoproteomics, SILAC, signaling, chronic and transient stress responses

## Abstract

The p38 members of the mitogen-activated protein kinases (MAPKs) family mediate various cellular responses to stress conditions, inflammatory signals, and differentiation factors. They are constitutively active in chronic inflammatory diseases and some cancers. The differences between their transient effects in response to signals and the chronic effect in diseases are not known. The family is composed of four isoforms, of which p38α seems to be abnormally activated in diseases. p38α and p38β are almost identical in sequence, structure, and biochemical and pharmacological properties, and the specific unique effects of each of them, if any, have not yet been revealed. This study aimed to reveal the specific effects induced by p38α and p38β, both when transiently activated in response to stress and when chronically active. This was achieved via large-scale proteomics and phosphoproteomics analyses using stable isotope labeling of two experimental systems: one, mouse embryonic fibroblasts (MEFs) deficient in each of these p38 kinases and harboring either an empty vector or vectors expressing p38α^WT^, p38β^WT^, or intrinsically active variants of these MAPKs; second, induction of transient stress by exposure of MEFs, p38α^−/−^, and p38β^−/−^ MEFs to anisomycin. Significant differences in the repertoire of the proteome and phosphoproteome between cells expressing active p38α and p38β suggest distinct roles for each kinase. Interestingly, in both cases, the constitutive activation induced adaptations of the cells to the chronic activity so that known substrates of p38 were downregulated. Within the dramatic effect of p38s on the proteome and phosphoproteome, some interesting affected phosphorylation sites were those found in cancer-associated p53 and Hspb1 (HSP27) proteins and in cytoskeleton-associated proteins. Among these, was the stronger direct phosphorylation by p38α of p53-Ser309, which was validated on the Ser315 in human p53. In summary, this study sheds new light on the differences between chronic and transient p38α and p38β signaling and on the specific targets of these two kinases.

## 1. Introduction

The MAPKs are a family of structurally homologous kinases with partially overlapping substrate repertoires, tissue distribution, and biological function. In mammals, there are four major MAPK subfamilies, ERK (ERK1, ERK2), JNK (JNK1-3), ERK5/BMK, and p38 (p38α, p38β, p38γ, and p38δ). These MAPKs regulate critical cellular processes, including proliferation, apoptosis, and embryogenesis [1,2,3,4]. Although each MAPK sub-family is activated via a distinct pathway, many of their activating stimuli affect more than one MAPK pathway. For example, numerous stress signals simultaneously activate both the JNK and p38 pathways [5,6].

The p38 proteins are predominantly involved with mediating the response to stress signals such as osmotic shock, radiations, and oxidative stress [6,7,8,9,10,11,12,13], as well as differentiation and inflammatory stimuli [14]. Strong activators of p38 are anisomycin, tunicamycin, hydroxyurea, and cycloheximide. Anisomycin, whose specific effects were studied here, is an inhibitor of protein synthesis produced by *Streptomyces* spp. The drug binds the 60S ribosomal subunit and blocks peptide bond formation [15,16,17,18]. Importantly, anisomycin-stimulated signaling already occurs at concentrations below those required for inhibiting translation (sub-inhibitory concentrations) [16].

p38α and p38β are highly similar, expressed in all tissues, and are sensitive to the same pharmacological inhibitors [19,20,21]. p38α is considered to be more important for signaling since it is expressed at higher levels than p38β, and p38α knockout mice fail to complete embryogenesis [22,23,24]. p38β knockout mice, on the other hand, are viable and fertile [25]. However, the essentiality of p38α is not a result of the low expression levels of p38β, as the expression of p38β from the p38α promoter cannot rescue the embryonal lethality of p38α knockout mice [23,24,26], implying that although p38α and p38β are highly similar, they have distinct functions. Furthermore, the different p38s are differentially activated by MKK3 and MKK6, resulting in specific downstream effects [27].

In response to stimuli, MAPKs are commonly activated transiently [6,7,8,9,10,11,12,13]. Continuous activation is abnormal and monitored in chronic inflammatory diseases and cancer [6,12]. It implies that the effects of constant MAPK activity are different to those of transient responses. The role of p38 in the etiology of these diseases is not clear, but as its activation per se in the liver is sufficient to impose some disease symptoms, it may play a causative role in fatty liver diseases [28] and other maladies [29,30,31,32,33]. Moreover, recent studies have shown that p38s play a role in the behavior and function of cancer stem cells [4]. Therefore, p38s and their substrates are considered to be potential drug targets in treatment of different maladies [4,6,12,34,35,36].

MAPKs are activated by dual phosphorylation of the Thr-X-Tyr (TXY) motif, located within the activation loop of their kinase domain. Dual phosphorylation is commonly catalyzed by upstream activating kinases, known as MAPK kinases (MAPKK). Under particular conditions or in some cell types, p38 can be activated via induced autophosphorylation [37].

The absolute dependence of MAPKs on upstream regulation for activation, particularly the fact that each MAPKK activates several MAPK isoforms, hinders the possibility of experimentally activating a given MAPK alone for the study of its specific physiological and pathological functions. This difficulty can be now circumvented with the use of intrinsically active (MAPKK-independent) variants of MAPKs. Such variants, carrying point mutations that render them capable of autophosphorylation/autoactivation in vitro and in cellulo, have been discovered for p38 and ERK [38,39,40,41,42].

This research focused on the complementarity and differences in the specific activities of p38α and p38β, as well as the transient versus chronic activation of these enzymes. This research took advantage of mouse embryonal fibroblast cell lines knocked out for p38α or p38β and harboring expression vectors for either the respective intrinsically active variants or native proteins. These cells were used for determining the specific chronic effects of each isoform per se on the cells’ proteomes and phosphoproteomes. To determine the rapid dynamic changes imposed by p38 activation on the proteomes and phosphoproteomes, the cells were treated with anisomycin. Interestingly, p38 activation resulted in the phosphorylation of critical molecules such as ERK, p53, and HSP27.

## 2. Results

The specific effects of continuous activation of p38α and p38β on the cell proteome and phosphoproteome were investigated here by combining proteomics and phosphoproteomics analyses using six different lines of MEFs labeled with light-, medium-, and heavy-isotope-labeled amino acids (lysine and arginine) (Appendix A): (1) Cells knocked out for p38α and harboring an empty plasmid were labeled with the medium isotope (Arg6 and Lys6). (2) Cells knocked out for p38α but stably expressing p38α^WT^ were labeled with the light isotope (Arg0 and Lys0). (3) Cells knocked out for p38α but stably expressing the constitutively active p38α^D176A+F327S^ were labeled with the heavy isotope (Arg10 and Lys8). (4) Cells knocked out for p38β and harboring an empty plasmid were labeled with the medium isotope (Arg6 and Lys6). (5) Cells knocked out for p38β but stably expressing p38β^WT^ were labeled with the light isotope (Arg0 and Lys0). (6) Cells knocked out for p38β but stably expressing the constitutively active p38β^D176A^ were labeled with the heavy isotope (Arg10 and Lys8). The cells transfected with the p38α variants were mixed and analyzed separately from the cells transfected with the p38β variants. The transfected p38s were tagged with the HA peptide sequence at their N-termini. The cultures were established via retroviral infections, and stably infected cells were sorted by FACS using a GFP reporter that was encoded in cis by the pMigR vector (Figure 1A). Notably, the cells’ growth rates were only marginally affected by the expression or by the knockout of the various p38 molecules (Figure 1B). Expression and phosphorylation of the TGY motif of the different p38s were validated by Western blot analysis (Figure 1C).

To monitor short-term, more direct effects of p38 activation, we studied the dynamic changes of the phosphoproteomes following exposure of cells to anisomycin. In this part of the study, wild-type MEFs were labeled with the light isotope (Arg0 and Lys0), and MEFs deficient in p38α (labeled with Arg10 and Lys8) or p38β (labeled with Arg6 and Lys6) were treated with 250 nM anisomycin for 0, 5, 10, 20, 30, and 60 min. Both the first part of the study (continuous activation) and the second part (dynamic activation) were analyzed via this SILAC methodology [43,44] based on the labeling scheme listed above, using tandem mass spectrometry (LC-MS/MS), as illustrated in Appendix A. The results of the first analysis revealed the effects of the continuous/chronic activity of the intrinsically active p38 variants, which were the consequences of both direct and indirect effects, while the results of the second dynamic analysis pointed to the more direct substrates of these p38s.

Large portions of the proteolyzed protein preparations were used for the enrichment of their phosphopeptides and subsequent phosphoproteome analyses, while small portions of each extract were kept for analysis of their proteomes (Appendix A). To increase the number of identified phosphopeptides, the tryptic peptides were first separated by strong cation exchange chromatography into five fractions using salt gradients. After acidification, the phosphopeptides in each fraction were enriched by two cycles of TiO_2_ affinity purification. The analyses were performed in two or three experimental repetitions, resulting in separate phosphoproteome and proteome datasets for each of the experiments (Appendix A). Growing the cells under similar growth conditions and mixing the light-, medium-, and heavy-isotope-labeled cell extracts just before the trypsin proteolysis and phosphopeptides enrichment enabled a reliable comparison of the relative LC-MS intensities of the identified peptides and phosphopeptides between the cell lines expressing the different p38 variants and the cells treated with the anisomycin. The raw LC-MS/MS data of each of the experiments were analyzed together by MaxQuant software [45], and the phosphoproteome results indicated the high reproducibility of the triplicate experiments, with Pearson correlations as high as 0.8 for the heavy-to-light-isotope (H/L) ratios of the different peptides (Appendix A).

### 2.1. Large Phosphoproteomes and Proteomes Were Obtained from MEFs Expressing Intrinsically Active Variants p38α or p38β or Treated with Anisomycin 

Altogether, the two parts of the study resulted in large numbers of identified phosphorylation sites, reaching 23,101 phosphosites derived from 4020 phosphoproteins. The phosphoproteome analyses of the cells expressing the intrinsically active p38α resulted in 12,085 phosphosites, and analysis of cells expressing the intrinsically active p38β resulted in 11,661 phosphosites being identified. In cells treated with anisomycin, 21,507 phosphosites were identified, with a large overlap in the identified sites between the experiments (Figure 2A). The identified sites were mostly phospho-Ser (84%) and phospho-Thr (15.5%), while only 0.5% of the identified sites were phospho-Tyr. These sites were mostly identified with high confidence localization probabilities of their phosphorylated amino acids [46] of at least 0.75 (Table 1 and Appendix A). In agreement with the Western blot analysis (Figure 1C), four phosphopeptides containing the TGY motif of the p38s were detected in the cells expressing intrinsically active p38 molecules, as expected [40,47,48] (Appendix A). The parallel analysis of the proteome datasets obtained from the same cell extracts included 2501 proteins from the lysates of the cells expressing the intrinsically active variants p38α and p38β. About 44% of these were represented among the observed phosphoproteome. In the lysates of the anisomycin-treated cells, 2792 proteins were identified, and 48% of these proteins were represented among the phosphoproteome (Appendix A).

### 2.2. Differential Modulation of the Phosphoproteomes by the Intrinsically Active p38α and p38β

Expressing each of the intrinsically active p38 variants affected differentially the phosphoproteomes of the MEFs. Expression of p38α^D176A+F327S^ in p38α^−/−^ MEFs affected about 38% of the identified phosphorylation sites relative to the p38α^−/−^ cells expressing p38α^WT^ (Figure 2B). In contrast, expression of p38β^D176A^ in p38β^−/−^ MEFs affected only about 15% of the identified phosphorylation sites relative to the p38β^−/−^ cells expressing p38β^WT^ (Figure 2B and Table 1). Importantly, the protein level of the p38α variants was lower than that of the p38β variants (Figure 1C). As many as 622 of these phosphosites were affected by the active variants of both p38α and p38β, indicating strong complementarity in the substrate repertoires of the two enzymes. On the other hand, these shared phosphorylation sites included 72 sites that were affected in the opposite directions by the intrinsically active p38 variants (Figure 2D).

The phosphorylation sites affected differentially by each p38 variant were selected using a *t*-test (with permutation-based FDR of 0.05) using the log2 of the ratios of the LC-MS signal intensities of the phosphopeptides [Log2(active variant/wild-type)] in the mutant p38 relative to the wild-type p38s (Appendix A). Of these identified phosphorylation sites, 2617 were either differentially phosphorylated up or down by the intrinsically active p38s, including 404 phosphosites derived from 86 known MAPK signaling pathway proteins (Figure 3A and Appendix A). These phosphopeptides were detected with no significant changes in their source proteins’ levels (Appendix A), meaning that these changes likely resulted from differential phosphorylations rather than from changes in their source proteins’ steady-state levels.

### 2.3. Differential Effects of p38α and p38β on the Stress-Induced Dynamics of the Phosphoproteome

To reveal the role of p38α and p38β in the cell response to anisomycin, the dynamic changes of the phosphoproteomes in the parental p38α^−/−^ and p38β^−/−^ MEFs following anisomycin treatment were studied (Appendix A). The most significant effects were observed at 5 min after exposure to the drug. About 68% of the phosphorylation sites were affected by more than 1.5-fold (Figure 2B), of which 1099 phosphorylation sites were differently affected between the parental p38α^−/−^ and p38β^−/−^ MEFs exposed to anisomycin throughout the entire time course. These sites are listed in Appendix A. Most of the phosphorylation sites were regulated similarly in all the cell lines, as indicated by hierarchical clustering (Appendix A). However, the anisomycin effects were more dramatic when p38β was absent. The fact that a large complementarity exists between the effects of the p38 isoforms, which regulate shared substrates (reviewed in [30]), was partially overcome by the dynamic phosphorylation analysis relative to the controls of the individual p38 knockouts. Interestingly, some sites were already different even in the untreated cells: Group 1 (Appendix A) included 17 phosphorylated sites elevated relative to the knockout cells at all the time points, and Group 2 (Appendix A) included 86 phosphorylation sites that were less phosphorylated in p38β^−/−^ cells and decreased by this stimulus in the other cell lines as well as in the p38β^−/−^ cells (Appendix A). Fourteen phosphorylation sites from Cluster #1045 were also differently changed in the intrinsically active variants experiments (Appendix A). In most of them, the active variant/wild-type phosphorylation ratios in p38β were higher than in p38α. These results point to a subset of phosphorylation sites that seems to be affected by p38β more than p38α.

### 2.4. Specific Effects of the p38s Activation on Known MAPK Substrates 

A subset of 30 proteins which are known substrates of p38 [50] was detected in both the phosphoproteomes of the MEFs expressing the intrinsically active p38s and in the anisomycin treatment (Table 2). This included ATF2, MSK1/2 (Rps6ka4), MAX, and cdc25, which are known substrates of p38 according to the KEGG database (pathway ID 04010) [51], and CERB and HSP27 (HSP1b), which are considered downstream substrates [52,53,54,55] (Appendix A). Reduction in phosphorylation levels relative to the wild type was observed in cells expressing intrinsically active p38 mutants in most of the affected known substrates of p38, including Thr51 and Thr53 of ATF2 and Ser343 and Ser347 of MSK1/2 (Figure 4A). These phosphorylations were observed mainly in cells expressing p38α^WT^ and were not elevated and were even downregulated by the intrinsically active p38α but less so in the presence of the intrinsically active p38β (Figure 3A). We reasoned that chronic activity of p38s, maintained by expression of the intrinsically active variants, leads to feedback inhibitory activities, leading to a reduction in the levels of phosphorylation in the p38 substrates (see Section 3).

In the anisomycin-treated phosphoproteome, 192 phosphorylation sites, derived from 67 proteins from the MAPK signaling pathway, were affected by the 5 min anisomycin treatment (Figure 2C). Generally, proteins of the MAPK signaling pathway (according to DAVID Bioinformatics Resources 6.8 [56,57]) were overrepresented at all time points for the anisomycin-treated phosphoproteomes (Appendix A). Interestingly, all these sites were affected to similar levels by the anisomycin treatment in the knockout cells. These are predicted to be phosphorylation substrates of p38 in the phosphoproteome databases Phospho.ELM [58], UniProt [59], and PhosphoSitePlus [60,61].

### 2.5. p38α and p38β Differently Affected HSP27 Phosphorylation

HSP27 is a known substrate of MAPKAPK2 (MK2) and MAPAPK3 (MK3) [62,63]. The HSP27-Ser86 phosphorylation site is conserved between humans and mice [59] and is known to be regulated by the p38-MK2 pathway, reviewed in [54]. Indeed, the phosphorylation of Ser86 of the murine HSP27 was mostly upregulated in MEF clones expressing p38α^D176A+F327S^ relative to the clones expressing p38α^WT^ (Figure 5A, Appendix A). This phosphorylation of HSP27 was also observed by Western blot analysis with antibodies directed against the (homologous) phospho-Ser82 of the human HSP27, confirming its upregulation in the presence of p38α (Figure 5C). Interestingly, both the Western blot and the proteomics analysis indicated a significant differential effect of the p38s on the HSP27 protein levels. The level of HSP27 expression in the p38α^−/−^ MEFs was very low (below the detection limit in this analysis) relative to in the p38β^−/−^ MEFs (Figure 5C and Appendix A). This observation suggests differential effects of the two variants of p38 MAPKs on the phosphorylations levels of HSP27 in the studied MEFs.

### 2.6. Differential Effects of the p38s on the Cytoskeleton

Many of the differentially modulated phosphorylation sites were observed in proteins associated with cancers and infections (Figure 3 and Appendix A). Among these, a significant group of 189 proteins, differentially phosphorylated by p38α relative to p38β, belonged to protein groups annotated with Structural Molecule Activity (GO:0005198), Cytoskeleton (GO:0005856), and Cell–Cell Adherent Junction (GO:0005913). These gene ontology annotations were enriched with a *p*-value lower than 0.01 (Appendix A) according to analysis with the DAVID Bioinformatics Resources 6.8 [56,57]. The p38 MAPKs are known to influence cytoskeletal components, such as F-actin [64]. To further investigate this issue, F-actin was stained in the different cell lines with the Phalloidin Conjugates TRITC and analyzed by confocal microscopy. The staining indicated that p38β^−/−^ MEFs have more elongated structures than p38α^−/−^ MEFs (Figure 6A–D). Additionally, expressing the constitutively active MAPKK MKK6^EE^ [30,65] resulted in a cytoskeleton morphology similar to that of p38α MEFs (in p38α^−/−^ MEF, Figure 6C,F). These results suggest that the activation of the other p38 isoforms, i.e., p38β, p38γ, or p38δ, present in these cells leads to the different structures of the cytoskeleton.

### 2.7. Differential Phosphorylation of Cancer-Related Proteins, including p53, by p38α and p38β 

An interesting group of proteins, grouped in ‘pathways in cancer’ (pathway ID 05200) in the KEGG database [51], was differentially phosphorylated by the intrinsically active p38s (Figure 3). Out of these 273 phosphorylation sites, in proteins that are involved in cancer development, 55 phosphosites in 34 proteins were differentially phosphorylated by p38α and p38β (Figure 3 and Appendix A), including the tumor suppressor p53 and APC, Raf1, Mdm2, MAX, and AKT1 (Figure 3B). Importantly, the affected phosphorylation sites of these proteins showed a significant number of interactions in STRING analysis [49] (Figure 3C).

Five phosphorylation sites were detected in p53, a tumor suppressor known to be affected by the p38 pathway [66]. Two of these sites were more strongly affected by the p38α relative to the p38β active variant (Figure 7A). The p53 302–314 peptide ALPTCSASpPPQKK was observed twice, with single and double phosphorylation on Ser307 and Ser309 residues. Ser309 of p53 was affected more strongly, as expected for a typical MAPK substrate motif (Figure 7A). To test whether p38s were directly responsible for these phosphorylations, an in vitro kinase assay was performed by incubating a fragment of the human p53 (94–360 aa) with different purified p38 variants and analysis of the p53 phosphorylation sites by trypsin digestion followed by LC-MS/MS. A phosphorylated Ser315 residue was detected on the human p53, which is homologous to Ser309 in the murine p53. These phosphorylation sites were detected only on p53 after incubation with p38^WT^, which was itself activated by constitutively active MKK6^EE^ (Figure 7C). These sites were not detected in p53 molecules incubated with the non-activated MKK6 control or in the presence of the intrinsically active p38 variants, which were less activated in vitro according to their phosphorylation levels (Figure 7B). The levels of phosphorylation of Ser309 of p53 protein decreased gradually following anisomycin treatment, while the doubly phosphorylated Ser-307-Ser309 first increased and then gradually decreased in the wild-type MEFs or in those devoid of one of the intrinsically active p38s. This observation raises the possibility that the two kinases can phosphorylate these two p53 sites (Figure 7D). Thus, we suggest that the chronic activation of the p38s leads to specific phosphorylation patterns of p53.

The expression of the active variants of p38 also affected differentially other p53-associated proteins. This observation was based on gene ontology annotation enrichment performed by the DAVID database (Appendix A) [56,57] using the differently changed phosphorylation sites, from which 15 proteins with the molecular function annotation of p53-binding proteins were selected, including the tumor suppressor p53-binding protein 1 (Trp53bp1) and nucleophosmin (Npm1). These 15 proteins had 144 phosphorylation sites (Appendix A and Appendix A) that were affected differentially by the expression of the active p38α or p38β variants (Appendix A).

## 3. Discussion

Distinguishing between the functions of highly similar enzyme isoforms is a complex task. This was achieved here for the almost identical p38α and p38β isoforms by combining cells knocked out for each isoform and expression of intrinsically active variants of each and by comparing the response of the knocked-out cells to anisomycin.

In this study, a large number of phosphorylation sites were discovered and quantified, pointing to strong complementarity in the repertoire of substrates affected by both p38α and p38β and to a smaller number of sites specifically modulated by each of the enzymes. In addition, many sites were differentially affected by transient activation relative to chronic activation of these p38 kinases. As expected, treating the cells with anisomycin had a much stronger effect on the general phosphoproteomes of the MEFs relative to the effect of replacing the endogenous p38s with their intrinsically active variants, likely because the cells respond to the drug by activating other pathways in addition to p38, for example, the JNK cascade, while the expression of intrinsically active variants results in strict exclusive activation of p38α or p38β. Different phosphorylation sites were modulated by the studied p38s by the intrinsically active p38 relative to those induced by the anisomycin treatment. These included proteins known to play a central role in carcinogenesis and inflammation. Several new or potentially interesting phosphorylation sites were discovered in these proteins, for example, p53, HSP27, Jun, and Rb1. Such a particularly interesting new p38 substrate site was identified in human p53-Ser315, which was validated as a direct substrate of p38 via in vitro kinase assay.

Our initial expectation was that the constitutive expression of the intrinsically active mutant p38s would increase the phosphorylation of the substrates of these MAPKs. However, a larger-than-expected portion of the phosphorylation sites was downregulated in cells expressing the intrinsically active p38s (for example, phosphorylation sites on ATF2 and Rps6ka4). We assume that this phenomenon was a consequence of negative feedback responses, which could be imposed by enhancing phosphatase activities, or by allosteric factors. The strong feedback implies that chronic p38 activity also induces cellular stress, as was learned from the effects of the active p38s on the cytoskeleton. These changes in the phosphoproteome are possibly interesting since they include many sites that are modulated in a fashion similar to chronic inflammation and cellular stresses. Analysis of the proteome and phosphoproteome of MEFs, in comparison to those of p38α^−/−^ and p38β^−/−^ MEFs, allowed the identification of particular anisomycin-induced responses that are specifically mediated by p38α and p38β. Prominent among the interesting proteins identified were ATF2, Arhgap1, Mylk, and Frmd4b. Likely, the early induced phosphorylation sites were direct substrates, while those changing late represented indirect events. Interestingly, anisomycin induced a stronger response in the phosphoproteomes of the p38β^−/−^ MEFs relative to in the p38α^−/−^ MEFs. It is important to note that the recombinant expression of the different p38 variants using a viral vector resulted in variable levels of expression, which may have affected the pattern of substrate phosphorylations, in addition to the specific activities of each enzyme. The selection of cells expressing similar levels of the enzyme using cell sorting based on the reporter green fluorescent protein helped to achieve better, uniform levels of expression of the p38 proteins in the cultured cells. Similar analyses of the effects of stress on the phosphoproteome were performed by [67] in response to anisomycin and by [68] in response to UV light, both in human osteosarcoma cultured cells, observing numerous changes in the phosphoproteomes induced by these stresses and identifying several stress-induced, p38-dependent phosphorylations. In addition, the substrates of p38α were studied more recently by following the phosphoproteome of mouse epithelial mammary tumor cells after knocking out this kinase or inhibiting it [55] or by following the effects of inhibiting p38 in human cultured endothelial cells in response to thrombin [69]. Furthermore, numerous direct substrates of the p38 kinases were recently identified using an in vitro scanning synthetic peptides assay [70,71] in which peptides containing the potential phosphorylation sites of about 300 different human kinases were followed while looking for the sites phosphorylated by each kinase separately. A few phosphorylation sites modified by p38 were also identified in mouse myoblasts by in vitro kinase assay [72]. In addition, interacting proteins of p38 were identified by a proximity labeling experiment, thus revealing potential substrates and regulated proteins [73]. Very useful and detailed databases of phosphorylation sites are available, and the phosphorylation sites discovered here were compared to the known mouse phosphoproteome listed sites in Phospho.ELM [58], UniProt [59], and PhosphoSitePlus [60,61]. The list of phosphorylation sites identified here and shared with those of [55] is indicated in Appendix A.

An interesting observation is the differential phosphorylations due to the presence of the intrinsically active expression of p38α or p38β. First, expression of the intrinsically active variant p38α^D176A+F327S^ affected the phosphoproteome more intensely than its counterpart p38β^D176A^. Second, the phosphorylation of HSP27-Ser82 was upregulated only in cells expressing the intrinsically active variant of p38α. HSP27 is known to maintain cytoskeletal integrity, in part via MK2 or MK5 [74], and to interact variably with actin filaments or with monomeric actin, influencing actin polymerization and de-polymerization [75,76,77]. This function of HSP27 can explain the dramatic difference (in 189 proteins) in the effect of p38α and p38β on cytoskeletal proteins and, consequently, on F-actin rearrangement (Figure 6). HSP27-Ser82 was also altered by the anisomycin treatment and reached its peak level after 10 min, while p38α was phosphorylated already at 5 min (Figure 5 and Appendix A), suggesting that HSP27-Ser86-Pi is likely a direct product of phosphorylation by p38.

A subset of cancer-related proteins was more strongly phosphorylated in cells expressing intrinsically active p38α than in cells expressing active p38β. Among these affected proteins were adenomatous polyposis coli protein (APC), histone deacetylase 2, BCL2-associated agonist of cell death, and transformation-related protein 53 (p53) [66,78]. In the current study, one of the p53 peptides was found to be phosphorylated on two residues, Ser307 and Ser309 (Figure 7). The phosphorylation site on Ser309 is predicted by the UniProt database as phosphoserine that can be phosphorylated by AURKA, CDK1, and CDK2. The phosphorylation sites on the mouse p53-Ser307 and Ser309 were identified in our datasets as differentially phosphorylated sites between the p38α and p38β datasets.

Human p53 is known to be phosphorylated by p38 on Ser15, Ser33, Ser46, and Ser392 but not on Ser315 and Ser313. The phosphorylation site on Ser315 (which is equivalent to Ser309 of the mouse p53) was already described as a phosphorylation site by other kinases, such as Aurora kinase, GSK3β, CDK1, and CDK9, but not by p38s (according to the UniProt database) [79,80]. Our in vitro kinase assay revealed that activated wild-type p38s could also phosphorylate this site directly. However, the in cellulo analysis showed that the intrinsically active p38α affected these phosphorylation sites more than the wild-type p38α, while no effect was observed due to the p38β variant expression. This difference can be explained by the other proteins present in the in vivo and not in the in vitro experiment. It is also possible that phosphatases present in the in vivo environment affected this phosphorylation site on p38α (Figure 7B,C).

The p38 MAP kinases and their signaling cascades have been studied extensively, mostly due to their role in mediating different stresses, inflammatory responses, and cancer [6,12,13]. As was emphasized here, cells exposed to stress for longer times adapt to the stress conditions by altering their proteomes and phosphoproteomes. We hope that this study has succeeded in shedding new light on specific substrates repertoires of p38α and p38β, in response to both transient and continuous activation, with potential implications for a better understanding of their role in disease progression and possibly even the design of new therapeutic approaches. The observation of many potential substrates of the studied p38s in proteins involved with the cytoskeleton, cell cycle, tumor suppression, and stress response may facilitate and encourage further research on these important issues. Continued activation of the MAPKs is observed in many inflammatory conditions and cancer, and the molecular mechanism of adaptation, primarily negative feedback mechanisms, may be strongly involved in disease etiology [81]. The findings obtained here may help design novel treatment strategies that take into consideration the feedback activity.

## 4. Materials and Methods

### 4.1. Cell Culture

Mouse embryonic fibroblasts (MEFs), p38α^KO^ and p38β^KO^ MEFs [82], and HEK293 cells were maintained in DMEM supplemented with 10% FBS, 1% L-glutamine, 1% Na pyruvate, 0.1% penicillin and streptomycin (all from Biological Industries, Beit Haemek, Israel). The cells were grown in a humidified incubator with 5% CO_2_ at 37 °C. Proliferation rates were calculated based on cell counting on consecutive days by TC20 Automated Cell Counter (Bio-Rad, Hercules, CA, USA) and methylene blue staining.

### 4.2. Infection and Transfection of Cultured Cells with Constitutively Active p38 Mutants

MEFs were transduced with retroviral vectors containing the cDNAs encoding p38α^WT^, p38β^WT^, constitutively active p38α^D176A+F327S^ or p38β^D176A^ [40,41,48], or with an ‘empty’ retroviral vector. The pMigR retroviral expression vector contained the EGFP ORF conjugated to an IRES element, serving as the transfection marker (a gift from Debbie Yablonski, Technion, Israel).

To produce the viral vectors, HEK-293 cells were co-transfected by the calcium phosphate method with the retroviral vector pMigr containing the indicated p38 variants and the packaging plasmid PCL-ECO in a 1:1 ratio. Virus-containing supernatants were collected at 48 and 72 h post transfection, filtered through a 45 µm filter, supplemented with polybrene at a final concentration of 8 µg/mL, and added to MEF cells plated 24 h before the procedure. Infection efficiencies were assessed by fluorescent microscopy or FACS analysis. The cells were then sorted using the FACSAria IIIU Cell Sorter (BD Biosciences, San Jose, CA, USA), and EGFP-positive cells were collected and expanded.

### 4.3. Anisomycin Treatment

MEFs were plated one day before treatment. Before treatment, the media containing the light, medium, and heavy amino acids were replaced with fresh media, with or without 150 nM anisomycin (Sigma, St. Louis, MO, USA). After 5 min of incubation with the anisomycin, the proteins were collected for phosphopeptides enrichment.

### 4.4. SILAC Labeling

For SILAC analysis [44], the intrinsically active cells transfected by variants p38α^D176A+F327S^ and p38β^D176A^ were labeled with the heavy forms of lysine (^13^C_6_,^15^N_2_-Lys: Lys8) and arginine (^13^C_6_,^15^N_4_-Arg: Arg10), the wild-type-p38α- and p38β-transfected cells were labeled with the light forms of lysine (^12^C_6_,^14^N_2_-Lys: Lys0) and arginine (^12^C_6_,^14^N_4_-Arg: Arg0), and the empty-vector-transfected MEFs were labeled with the medium form of lysine (^13^C_6_,^14^N_2_-Lys: Lys6) and arginine (^13^C_6_,^14^N_4_-Arg: Arg6) (Cambridge Isotope Laboratories, Andover, MA, USA).

### 4.5. Proteins Extraction and Phosphatase Inhibition

The SILAC-labeled cells were grown in 15 cm culture plates, and 3 mg of each light, medium, and heavy amino acid labeled protein extract was mixed for proteomics analysis and phosphopeptides enrichment (separate to the transfected p38α and the p38β variants). The proteins from the different cell lines were extracted from the plates with lysis buffer composed of 8 M urea, 75 mM NaCl, 1 mM NaF, 50 mM Tris pH 8, 1 mM β-glycerolphosphate, 1 mM Na orthovanadate, 1 mM Na pyrophosphate, protease inhibitor cocktail (1:200, Sigma-Aldrich, St. Louis, MO, USA), and 1 mM PMSF.

### 4.6. In-Solution Protein Digestion

Before tryptic digestion, disulfides were reduced by incubation with 2.8 mM DTT (Sigma) for 30 min at 60 °C and carbamidomethylated with 8.8 mM iodoacetamide (Sigma) for 30 min at room temperature in the dark. Trypsin (sequencing grade, modified, Promega, Madison, WI, USA) was added in a ratio of 1:50 to the proteins and incubated overnight at 37 °C. A second dose of trypsin was added in the morning and incubated for four more hours. The samples were then acidified with 0.1% TFA and desalted with Sep-Pak C18 reversed-phase cartridges (Waters, Milford, MA, USA).

### 4.7. Fractionation of Peptides by Strong Cation Exchange Chromatography

Peptides were dissolved in 0.1% TFA and loaded on a 1 mL Resource S Column (GE Healthcare, Uppsala, Sweden) on a Jasco LC-Net II/ADC HPLC. Peptides were eluted with a KCl gradient from 0 to 350 mM containing 7 mM KH_2_PO_4_ (pH 2.65) and 30% acetonitrile over 92 min at a flow rate of 0.5 mL/min. The elution was divided into 10 fractions, which were then affinity enriched for phosphopeptides.

### 4.8. Phosphopeptides Enrichment on Titanium Beads (MOAC)

A 5 mg amount of Titansphere chromatography material (TiO_2_) (GL Sciences, Tokyo, Japan) was added to a 100 µL loading solution (30 mg/mL 2,5-dihydroxybenzoic acid (Sigma-Aldrich) in 80% acetonitrile and 0.1% TFA) and incubated with shaking for 10 min at room temperature. Loading solution was added to the fractions in a ratio of 1:6. Then, the TiO_2_ beads were transferred to the samples and shaken for 30 min at room temperature. The beads were washed with 1 mL washing solution #1 (30% acetonitrile, 3% TFA) and 1 mL washing solution #2 (80% acetonitrile, 0.1% TFA). The beads were transferred to C8 Stage-Tips [83], and the peptides were eluted with 100 µL elution buffer #1 (20% acetonitrile, 325 mM NH_4_OH) followed by 100 µL elution buffer #2 (40% acetonitrile, 325 mM NH_4_OH) combined into one tube. The eluate was dried in a vacuum centrifuge (LABCONCO, Kansas City, MO, USA) and desalted on C18 Stage-Tips.

### 4.9. Peptides Analysis by LC-MS/MS

Mass spectrometry analysis was performed on a Q Exactive Plus Mass Spectrometer (Thermo-Fisher Scientific, Waltham, MA, USA) using the positive ion mode and full MS scan repetitively followed by higher-energy collision dissociation (HCD) of the 10 most dominant ions selected from the full MS scans. Reversed-phase chromatography was performed with home-made, about 30 cm long, 75 μm inner diameter capillaries packed with 3.5 μm silica ReproSil-Pur C18-AQ resin (Dr. Maisch GmbH, Ammerbuch-Entringen, Germany), as previously described [84]. Peptides were eluted using a linear gradient of 5–28% acetonitrile in 0.1% formic acid at a flow rate of 0.15 μL/min for 3 h. Full-scan MS spectra were acquired at a resolution of 70,000 at 200 *m*/*z* with a target value of 3 × 10^6^ ions. Fragmented masses were accumulated to an automatic gain control (AGC) target value of 10^5^ with a maximum injection time of 100 ms. Fragmentation was performed for charge states of 2 to 7. The peptide match option was set to Preferred. The normalized collision energy was set to 25%, and MS/MS resolution was 17,500 at 200 *m*/*z*. Fragmented *m*/*z* values were dynamically excluded from further selection for 20 s.

### 4.10. LC-MS/MS Data Analysis

The MS data were interpreted by MaxQuant software version 1.5.0.25 [45], and the database was searched with the Andromeda search engines [85], searching against the mouse section of the UniProt database from July 2014 [59] containing 51,547 entries, including the human p38 proteins and their mutated variants and human p53. A mass tolerance of 4.5 ppm for the precursor masses and 20 ppm for the fragments was accepted. Proteins identification and phosphorylation sites were filtered to a 0.01 false discovery have rate (1% FDR), followed by statistical analysis with Perseus version 1.5.6.0 [86]. Phosphorylation sites that were differentially modulated by the presence or absence of the p38s were selected by ANOVA multi-sample tests at all the time points of each cell line separately.

### 4.11. Western Blot Analysis

Western blot analyses were performed with the following antibodies: anti-HA (hybridoma clone 12CA5, ATCC); p38α (sc-535) and p38β (sc-390984) (Santa Cruz Biotechnology, Santa Cruz, CA, USA); phospho-p38 MAP kinase (Thr180/Tyr182) (Cell Signaling, Danvers, MA, USA); actin (MP Biomedicals, Irvine, CA, USA). The immune detection was performed by peroxidase-conjugated secondary antibodies (goat anti-mouse; goat anti-rabbit; Jackson Immunoresearch, West Grove, PA, USA), and signals were developed using WesternBright ECL (Advansta, San Jose, CA, USA).

### 4.12. Protein Expression in Bacteria

For bacterial expression of the p38 proteins, the pET15b and pET28a vectors N-terminally tagged with hexahistidine were used. Protein expression was induced in the *Escherichia coli* Rosetta strain (Novagen, Madison, WI, USA). For expression of p53, we used a pREST plasmid (Invitrogen, Carlsbad, CA, USA) encoding a fusion protein modified in frame with N-terminal hexahistidine and 94–360 residues of the human p53, followed by the B. stearothermophilus dihydrolipoyl reductase sequence (2–85 aa). This plasmid was constructed in the laboratory of Alan Fersht, Cambridge, UK, and kindly provided to us by Gad Shuster at the Faculty of Biology, Technion, Israel. The 6xHis-p53 protein was purified from *E. coli* using metal affinity chromatography with Ni–nitrilotriacetic acid (Ni-NTA)-loaded beads (Adar Biotech, Yavne, Israel). Protein concentrations were determined by the Bradford method.

### 4.13. In Vitro Kinase Assays

Purified p38 proteins were incubated with an active mutant of MKK6 in which both Ser207 and Thr211 were mutated to Glu (termed MKK6^EE^) to activate the different p38s. To initialize the reaction, 45 μL of the reaction mixture containing 20 mM MgCl_2_, 20 mM 2-glycerolphosphate, 0.1 mM Na_3_VO_4_, 1 mM dithiothreitol (DTT), 50 μM ATP, 25 mM HEPES, and 40 μg of p53, pH 7.5, was added to 5 μL of activated p38 enzyme (0.2 μg of purified p38) for 10 min at 30 °C, with agitation, and were terminated by transfer to ice. Next, 5 μg total protein of the reaction mix, containing the wild-type p38α (p38α^WT^), p38α^D176A+F327S^, wild-type p38β (p38β^WT^), or p38β^D176A^ proteins and p53 were digested with trypsin to detect phosphorylation on p53 by LC-MS/MS analysis.

### 4.14. Confocal Microscopy

MEFs, transfected with the four p38 variants, were grown on sterile coverslips in 6-well plates. Cells were washed with PBS with 100 mg/L calcium chloride and 100 mg/L magnesium chloride three times, fixed with 4% paraformaldehyde for 15 min, and washed again with PBS three times. Next, the cells were perforated with 0.1% Triton X100 for 30 min. After washing the cells three times with PBS, they were stained with 3.5 μM Phalloidin Conjugates TRITC (Sigma, St. Louis, MO, USA) and 1:1000 Hoechst for 60 min. The slides were stored in PBS at 4 °C until use. The confocal images were taken by Confocal LSM 510 META Microscope (Zeiss, Oberkochen, Germany).

## Figures and Tables

**Figure 1 ijms-24-12442-f001:**
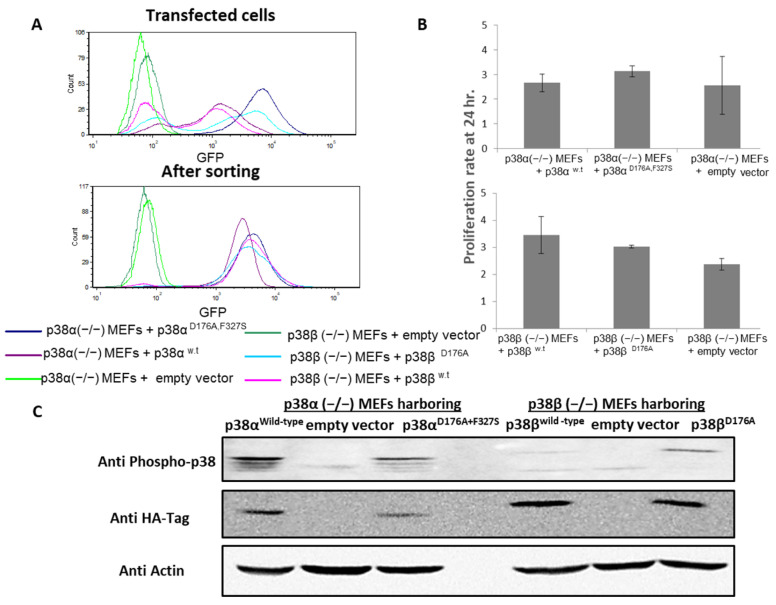
Establishment and characterization of MEF clones expressing the recombinant p38 molecules. (**A**) Sorting of MEFs expressing recombinant p38 variants from the infected MEFs population using the GFP marker expressed within the same vector as the p38 variants. Histograms of pre-sort and post-sorts are shown; (**B**) proliferation rates of the MEF cell lines according to methylene blue staining; (**C**) cell extracts prepared from the indicated clones were analyzed by Western blotting with anti-phospho-p38, anti-HA antibodies, and anti-actin, which served as a gel loading control.

**Figure 2 ijms-24-12442-f002:**
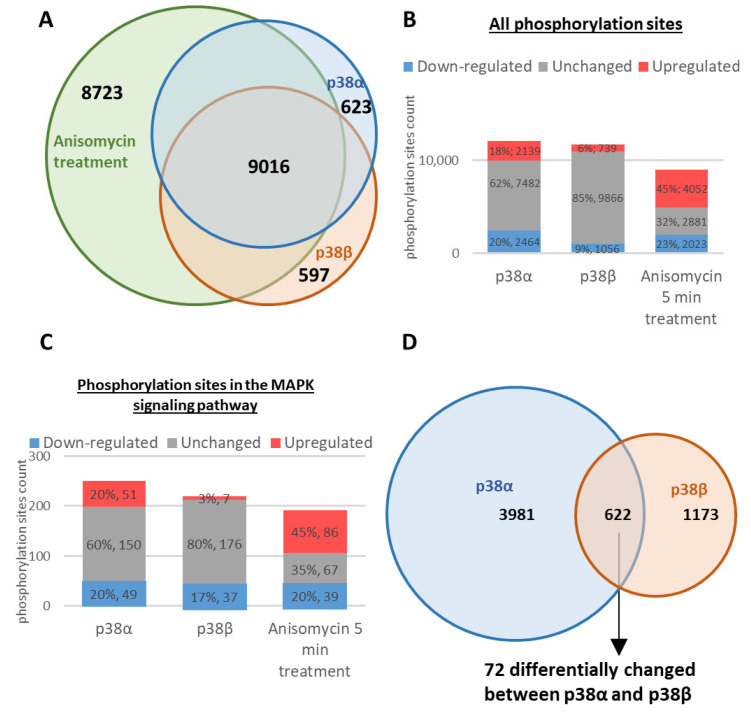
Larger numbers of phosphosites annotated to the MAPK signaling pathway were affected by the anisomycin treatment and by the expression of the constitutively active variants of p38α and p38β. (**A**) Identified phosphorylation sites in cells expressing p38α or p38β molecules and in anisomycin-treated cells (Appendix A); (**B**) the anisomycin treatment induced larger changes in the numbers and levels of phosphorylations. Phosphorylation sites were those affected by more than 1.5-fold due to the presence of the active variants or due to the anisomycin treatment at 5 min; (**C**) these phosphorylation sites included many sites in proteins belonging to the MAPK signaling pathway; (**D**) phosphosites affected in the p38α and p38β experiments by more than 1.5-fold relative to the wild-type variants of p38α and p38β, including 72 sites that changed in an opposite direction.

**Figure 3 ijms-24-12442-f003:**
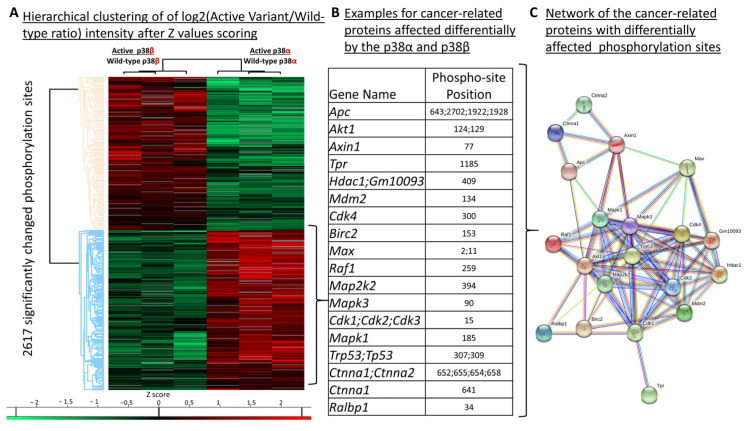
Intrinsically active variants of p38α and p38β have different effects on the MEF’s phosphoproteomes. (**A**) Heatmap of significantly changed phosphorylation sites in cells expressing the p38α or p38β molecules; (**B**) examples of cancer-related proteins whose phosphorylation levels were affected differentially by the p38α and p38β in the second cluster in panel A (pale blue). The identified phosphorylation sites’ positions are listed; (**C**) network of the cancer-related proteins with differential patterns of phosphorylation levels (created by the STRING website [49]). The phosphorylation sites of these proteins were upregulated due to the expression of p38α^D176A+F327S^. The network analysis indicated a large number of interactions between these proteins and different types of interaction evidence (colored lines).

**Figure 4 ijms-24-12442-f004:**
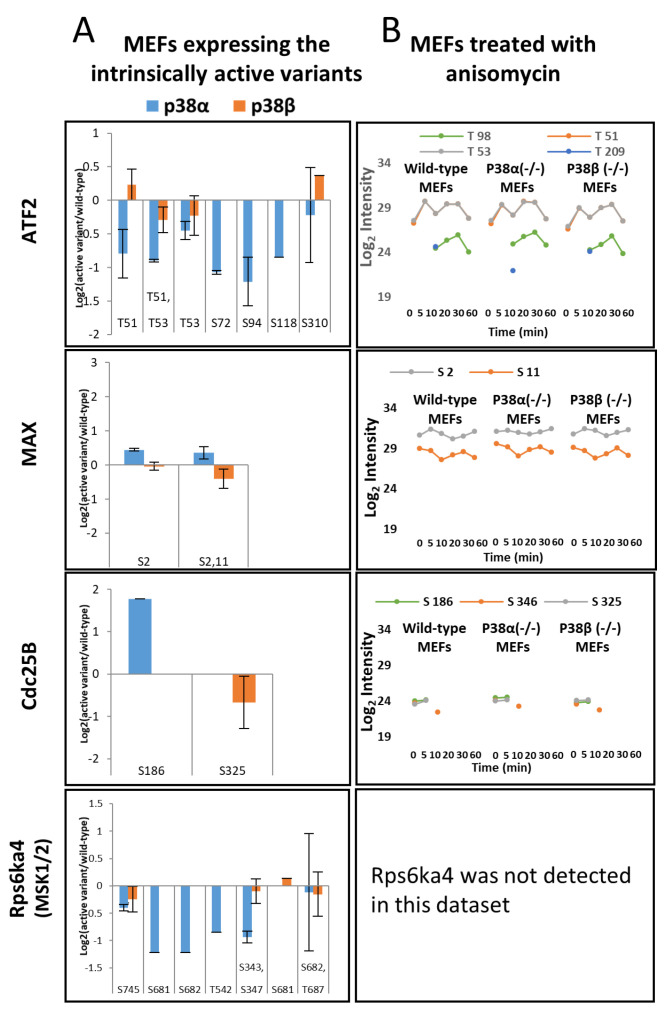
Examples of the effects of expressing the active p38 variants and of anisomycin treatment on known substrates of p38: ATF2, HSP27 (HSPb1), Rps6ka4 (MSK1/2, MAX, and Cdc25B. (**A**) The bar charts represent the Log2 of the ratio of the active variant/wild type of the listed p38 substrates in the p38α (in orange) and in the p38β (in blue) experiments; (**B**) profile plots of phosphorylation sites of p38 substrates at five time points after the anisomycin treatment. The missing data points in the profile plots indicate phosphosites not detected by the LC-MS/MS analysis.

**Figure 5 ijms-24-12442-f005:**
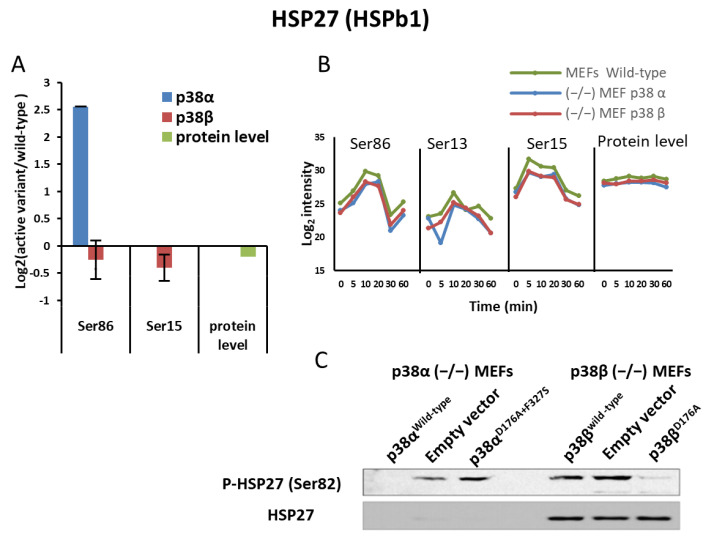
Expression and phosphorylation levels of HSP27 were differentially affected by the expression of active variants of p38α and p38β. (**A**) The bar charts represent the Log2 of the ratio of HSP27 phosphorylation and protein levels in cells expressing the active variant/wild type of HSP27 of p38α (orange) or p38β (blue) and the ratio of protein levels in green; (**B**) profile plots of phosphorylation site levels of HSP27 and its protein levels at five different time points after the anisomycin treatment; (**C**) cell extracts were separated by 12.5% SDS-PAGE, blotted to nitrocellulose, and ECL signal was developed with anti-phospho-HSP27 (Ser82) or anti-HSP27 and peroxidase-conjugated secondary antibodies.

**Figure 6 ijms-24-12442-f006:**
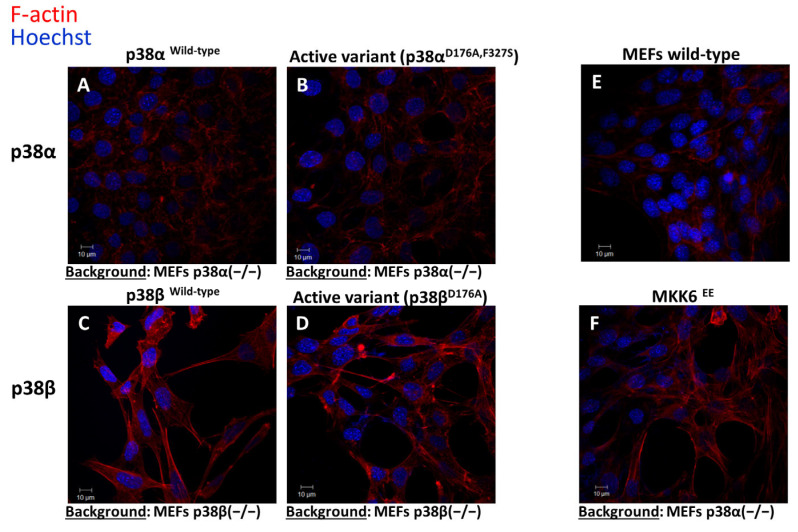
Expression of the intrinsically active variants of p38α and p38β affects cellular cytoskeleton and morphology. (**A**) p38α^−/−^ cells expressing p38α^WT^; (**B**) p38α^−/−^ cells expressing p38α^D176A+F327S^; (**C**) p38β^−/−^ cells expressing 38β^WT^; (**D**) p38β^−/−^ cells expressing the 38β^D176A^; (**E**) wild-type MEFs; (**F**) p38α^−/−^ cells expressing MKK6^EE^.

**Figure 7 ijms-24-12442-f007:**
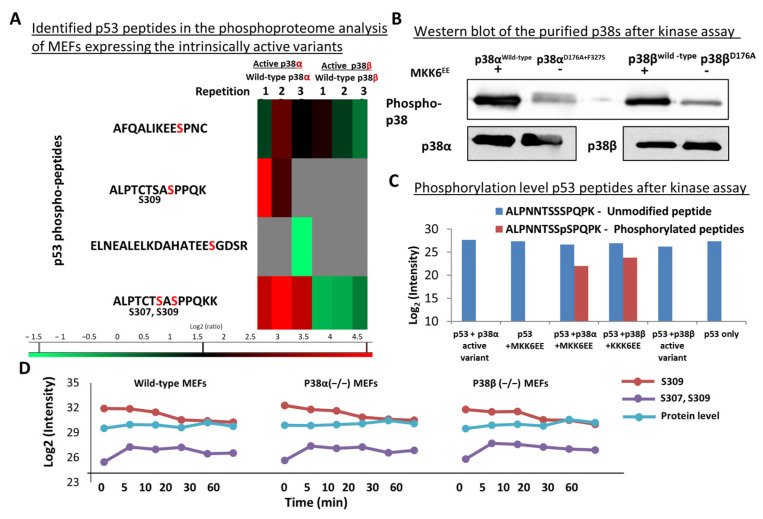
p53 was phosphorylated in vitro by p38α and p38β at the same site as observed in the phosphoproteome analysis. (**A**) Heatmap of Log2 (active variant/wild type) of p53 phosphopeptides. The amino acids labeled in red are the phosphorylated sites. (**B**) the intrinsically active p38 mutants were active, as indicated by their phosphorylation levels. The purified p38s, after kinase assay, were separated by 10% SDS-PAGE, Western blotted, and developed with anti-phospho-p38, anti-p38α, and anti-p38β antibodies; (**C**) signal intensity levels of the Ser315 phosphorylated and unmodified human p53 peptides; (**D**) profile plots of phosphorylation sites and protein levels of p53 at five different time points after the anisomycin treatment.

**Table 1 ijms-24-12442-t001:** Numbers of identified phosphosites in the different experiments.

	Total Phosphorylation Sites	Regulated Sites
Intrinsically active p38α experiment	12,085	
		Upregulated 2139
		Downregulated 2464
Intrinsically active p38β experiment	11,661	
		Upregulated 739
		Downregulated 1056
Anisomycin treatment	21,507	
		Upregulated 663
		Downregulated 436
Total	23,101	

**Table 2 ijms-24-12442-t002:** Table of the p38 known substrates and phosphosite positions detected in the phosphoproteomes datasets *.

Gene Names	Phosphosite Positions	Intrinsically Active Variants	Anisomycin Treatment
*Atf2*	51, 53, 72, 149, 94, 118, 310, 212, 191, 150, 118, 98, 209, 44	+	+
*Foxo3*	7, 12, 279, 252, 424, 283, 299, 298, 285	+	+
*Jun*	62, 63	+	+
*Mef2a*	98, 108, 479, 464, 400	+	+
*Mef2c*	222	+	+
*Mitf*	237	+	+
*Trp53*; *Tp53*	307, 309, 375, 350, 353	+	+
*Stat1*	727		+
*Cdt1*	165, 164, 114, 403, 414		+
*H2afx*	121, 124		+
*Rb1*	31, 243, 364, 599, 605, 367, 601, 617, 800, 814, 819, 816	+	+
*Rbm17*	155, 169, 222, 224	+	+
*Gsk3a*; *Gsk3b*	215, 216	+	+
*Mapkapk2*	208, 211		+
*Rps6ka4*	745, 347, 681, 682, 687, 771, 542, 773, 678	+	+
*Casp3*	12, 26	+	+
*Casp8*	188, 213	+	+
*Cdc25b*	186, 325, 346	+	+
*Ccnd3*	68	+	+
*Gys1*	589, 593, 608, 588	+	+
*Spag9*	569, 566, 53, 39, 150, 21, 201, 151, 19, 153, 61, 62, 80, 568, 185, 107, 97, 113, 97, 113	+	+
*Cdkn1c*	6		+
*Pip4k2b*	326, 322		+
*Psmd1*	315, 311, 273, 270	+	+
*Tab1*	7	+	+
*Egfr*	695, 697, 993		+
*Adam17*	794		+
*Zfyve20*	218, 216, 21, 229, 214, 225, 213, 635, 642	+	+
*Hspb1*	86, 13, 15	+	+

* Some of these phosphorylation sites were identified in only one of the experiments, and these are indicated in Appendix A.

## Data Availability

Data are available in a public, open-access repository. The data were deposited to the Pride through the ProteomeXchange in project accession: PXD038389.

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
