# Peer review of "Differential Modulation of the Phosphoproteome by the MAP Kinases Isoforms p38α and p38β"

_ijms, 2023, doi:10.3390/ijms241512442_

Round 1
Reviewer 1 Report
This manuscript by Melamed-Kadosh et al. presents an interesting study that investigates a significant and important issue, namely, the differences between chronic and transient activation of p38α and p38β in the phosphorylation of their specific protein targets. The paper presents an effective mix of experimental approaches to attack the question at hand. They combine large-scale proteomics and phosphoproteomics analyses of MEF cells where one of the p38 isoforms was knocked out and constitutively active variants were introduced to mimic chronic activation of the specific isoform, with studies on transient activation using anisomycin as a stress activator. This approach led to several novel and important findings. First, distinct differences were observed between the two almost similar p38 isoforms in both the proteome and phosphoproteome analysis, suggesting that their roles and functions are non-overlapping. Second, a strong negative feedback mechanism was observed in the cells expressing the intrinsically active p38 mutants, demonstrating that constitutive activation of the p38 isoforms leads to cellular adaptation by downregulation of potential p38 phosphorylation targets. Third, specific phosphorylation target sites that were subject to differential phosphorylation by the two p38 isoforms were identified on proteins involved in central cellular pathways and in cancer, including p53 (where Ser315 on human p53 was shown to undergo p38a-mediated phosphorylation), HSP27, APC, and more. These findings provide novel data on the mechanism of action of p38 isoforms and the differences between them, which may contribute to the development of future treatments.
The experimental design is adequate, the manuscript is well written and the results are compared to the related literature. I only have a few minor remarks that may help to improve the manuscript:
(1) The SILAC experiments could benefit from both more detail and a simple explanation of the experimental design, both in the text and under Materials and Methods, so that it would be clearer to the non-expert readership. For example, it is not clear which cells received which isotopes.
(2) Regarding specific findings, such as the novel p38a phosphorylation site on Ser315 of human p53, or the specific alterations in cytoskeletal proteins: The reader would benefit from suggestion of the potential implications of these changes, even if only at the level of potential explanations at this point.
Author Response
Response to the Reviewers’ comments: We thank the reviewers for their efforts and useful suggestions, which we fully accept. We modified the text accordingly and included a file with the changes tracked. We believe that the manuscript is much improved by these comments.
Reviewer #1
Comments and Suggestions for Authors
This manuscript by Melamed-Kadosh et al. presents an interesting study that investigates a significant and important issue, namely, the differences between chronic and transient activation of p38α and p38β in the phosphorylation of their specific protein targets. The paper presents an effective mix of experimental approaches to attack the question at hand. They combine large-scale proteomics and phosphoproteomics analyses of MEF cells where one of the p38 isoforms was knocked out and constitutively active variants were introduced to mimic chronic activation of the specific isoform, with studies on transient activation using anisomycin as a stress activator. This approach led to several novel and important findings. First, distinct differences were observed between the two almost similar p38 isoforms in both the proteome and phosphoproteome analysis, suggesting that their roles and functions are non-overlapping. Second, a strong negative feedback mechanism was observed in the cells expressing the intrinsically active p38 mutants, demonstrating that constitutive activation of the p38 isoforms leads to cellular adaptation by downregulation of potential p38 phosphorylation targets. Third, specific phosphorylation target sites that were subject to differential phosphorylation by the two p38 isoforms were identified on proteins involved in central cellular pathways and in cancer, including p53 (where Ser315 on human p53 was shown to undergo p38a-mediated phosphorylation), HSP27, APC, and more. These findings provide novel data on the mechanism of action of p38 isoforms and the differences between them, which may contribute to the development of future treatments.
The experimental design is adequate, the manuscript is well written and the results are compared to the related literature. I only have a few minor remarks that may help to improve the manuscript:
(1) The SILAC experiments could benefit from both more detail and a simple explanation of the experimental design, both in the text and under Materials and Methods, so that it would be clearer to the non-expert readership. For example, it is not clear which cells received which isotopes.
Response: We thank the Reviewer for the helpful comments. Accordingly, we modified the text both in the Results and in the Materials and Methods and elaborated on the isotope labeling scheme of the different cells used for the SILAC experiment to make it clearer to non-expert readers (lines 99-125).
(2) Regarding specific findings, such as the novel p38a phosphorylation site on Ser315 of human p53, or the specific alterations in cytoskeletal proteins: The reader would benefit from suggestion of the potential implications of these changes, even if only at the level of potential explanations at this point.
Response: We thank the Reviewer for the insightful comments, and we elaborated on the potential implication in the discussion (lines 459-472). We direct the readers to excellent reviews in reference [6, 12,13] which explain very thoroughly the role of the p38 MAPK in diseases, response to stress, and cell cycle progression.
Reviewer 2 Report
Summary and overall comments:
Understanding the differences in the specificity of substrates of various p38 isoforms can lend insights into cellular biology. By using constitutively active variants of p38a and p38b as well as anisomycin treatment in knockout cells, authors attempted to study differences in p38a and p38b. Authors identified several differences in constitutively active variants of p38a and p38b. However, the response to anisomycin was highly similar between the wild type and knockout MEFs.
The main takeaways that “differential modulation of the phosphoproteome by the p38a and p38b” are inconclusive. The anisomycin treatment data indicates high level of complementarity between the p38 isoforms. While there were huge phosphoproteome differences between constitutively active variants of p38a and p38b, these differences might come from different protein expression levels of the variants and/or differential specificity of constitutively active variant, each of which requires additional experiments with appropriate controls. Despite of this limitation, the study highlights several p38 substrates and confirms previous findings – making critical contribution to the literature. The statements regarding differential phosphorylation could be altered to highlight this limitation. Additionally, the manuscript could mainly focus on similarity/complementarity between the p38a and p38b isoforms.
Major comments:
1. It looks like 6 different MEF cell lines were used: 1) Cells knockout for p38α and harbor an empty plasmid. 2) Cells knockout for p38α, but stably expressing p38αWT. 3) Cells knockout for p38α, but stably expressing the constitutively active p38αD176A+F327S. 4) Cells knockout for p38β and harbor an empty plasmid. 5) Cells knockout for p38β, but stably expressing p38βWT. 6) Cells knockout for p38β, but stably expressing the constitutively active p38βD176A. Can you confirm if the cells knockout for p38α harboring an empty plasmid were also knockout for p38β, and vice versa? Otherwise, all of the experiments are missing this double knockout control, which is essential for the manuscript. If the cells were not double knockout, can you measure expression level of endogenous p38β in p38α knockout cells, and vice versa. Expression of mouse p38α in p38β conditions (4,5,6) would complicate the interpretation.
2. Expression levels of constitutively active variants of p38α and p38β should be similar between the cell lines to make accurate and meaningful comparisons. Based on Figure 1C, it appears that protein levels of p38β are different from p38α. Differential levels of active variants can lead to differences in phosphoproteome.
a. There are substantially higher number of phosphosites that are differentially regulated for p38α active variant compared to p38β active variant. Were the protein expression levels different between these variants?
3. The phosphoproteomics data from anisomycin treatment experiment (Figure S1c) is in stark contrast to that from constitutively active variants of p38α and p38β (Figure S1a-b). As shown in Figure S7, temporal profile of majority of sites is similar if not identical between wild type MEFs, p38α (-/-) and p38β (-/-) suggesting minimal effects and differences between p38α and p38β to anisomycin treatment. As discussed in line 211, large complementarity exists among the p38 variants. This further raises the need for double knockout control or all p38 variant knockout control to truly study the specificity differences between the p38 variants as background expression of other variants complicates the interpretation.
a. Sites found in Cluster #1045 are interesting since they are differentially regulated in p38β (-/-) suggesting that these sites may be regulated by p38β. How many of these sites are also differentially regulated in the constitutively active variant experiment? Does the profile match between the two datasets for these sites?
b. These data also raises a concern that the constitutively active variants may have different specificity for substrates compared to wild-type proteins?
4. Please mention how upregulated, downregulated and differentially expressed phosphosites were determined in Figure 2 i.e. fold change and/or p-value cutoffs. It’s very confusing which cutoff are used where.
5. It’s unclear what’s shown in Figure 2D. The figure is not discussed in the results section either.
6. Section 2.4: p38a and p38b differently affect HSP27 phosphorylation and expression levels: this conclusion may only be partially correct. While constitutively active variant of p38a leads to higher phosphorylation of HSP27 and that HSP27 has low protein levels in p38a (-/-) cells, the temporal profile in anisomycin treatment data (Figure 5b) shows that large complementarity exists. Even though p38a or p38b are absent, HSP27 has similar phosphorylation and protein level profile, essentially contradicting the main statement.
7. Section 2.6: Again here with p53: although constitutively active variant shows differential phosphorylation, in vitro kinase assay suggest that both p38a and p38b can phosphorylate p53 at similar levels. How does the profile look in anisomycin treatment data for p53? These data suggest that there may not be differential phosphorylation of p53 by p38a or p38b.
Minor comments:
1. Figure 2B: Spelling error for y-axis label – should be phosphorylation site count
2. Figure 2C: Spelling error for title – should be signaling
3. It is challenging to visualize data in Figure 4 left panels because of overlapping labels with the bar plots. Make sure to place the labels appropriately.
4. Figure 5a is missing y-axis label.
5. Figures are not discussed chronologically. For example, Figure 5 is discussed in line 187 before Figure 4.
6. Please provide color bar for Figure S7. Also what does the heatmap represent – fold change, log2 intensity?
Author Response
Response to the Reviewers’ comments: We thank the reviewers for their efforts and useful suggestions, which we fully accept. We modified the text accordingly and included a file with the changes tracked. We believe that the manuscript is much improved by these comments.
Reviewer 2 Comments and Suggestions for Authors
Summary and overall comments:
Understanding the differences in the specificity of substrates of various p38 isoforms can lend insights into cellular biology. By using constitutively active variants of p38a and p38b as well as anisomycin treatment in knockout cells, authors attempted to study differences in p38a and p38b. Authors identified several differences in constitutively active variants of p38a and p38b. However, the response to anisomycin was highly similar between the wild type and knockout MEFs.
The main takeaways that “differential modulation of the phosphoproteome by the p38a and p38b” are inconclusive. The anisomycin treatment data indicates high level of complementarity between the p38 isoforms. While there were huge phosphoproteome differences between constitutively active variants of p38a and p38b, these differences might come from different protein expression levels of the variants and/or differential specificity of constitutively active variant, each of which requires additional experiments with appropriate controls. Despite of this limitation, the study highlights several p38 substrates and confirms previous findings – making critical contribution to the literature. The statements regarding differential phosphorylation could be altered to highlight this limitation. Additionally, the manuscript could mainly focus on similarity/complementarity between the p38a and p38b isoforms.
Response: The reviewer is correct in pointing to a common caveat with such studies: Indeed, similarly to other signaling cascades driven by several isoforms of the same kinases' family, the two p38s isoforms studied here are activated by shared upstream regulators and share many downstream substrates. We added a note about this issue in lines 86-88, 190-191, 373-375. In addition, since feedback regulatory loops are active in each of their pathways, the activation or inhibition of one of them affects the pattern of downstream phosphorylation of the other, further complicating the analysis of data obtained. Yet, this is the limitation of studying such intricate biological systems. In addition, the reviewer is also correct when pointing out the fact that these kinases were expressed recombinantly by transfection with a viral vector, including the wild-type protein, whose levels of expression were different in the different cultured cells, probably affecting their activity levels and patterns of detected phosphorylation above the technical background detection limits. Another common limitation that is very difficult to control, but certainly taken into consideration in data analysis. The specific effects of the anisomycin treatment on each of the p38s were indeed small relative to the larger specific effects of the continuous activation of the intrinsically active p38 variants. In retrospect, such results were somewhat expected since the anisomycin induces a strong stress response that is mediated also by other enzymes, including other kinases, while the chronic activation of the p38 variants induces feedback regulatory loops and stronger amplification of downstream cascades. To distinguish between the few direct individual effects of the p38s, we mainly relied on the anisomycin treatment, and to follow the individual long-term effects of their activation, we used the intrinsically active mutants, accepting the limitation that these also induce many indirect effects both upstream and downstream.
Major comments:
- It looks like 6 different MEF cell lines were used: 1) Cells knockout for p38α and harbor an empty plasmid. 2) Cells knockout for p38α, but stably expressing p38αWT. 3) Cells knockout for p38α, but stably expressing the constitutively active p38αD176A+F327S. 4) Cells knockout for p38β and harbor an empty plasmid. 5) Cells knockout for p38β, but stably expressing p38βWT. 6) Cells knockout for p38β, but stably expressing the constitutively active p38βD176A. Can you confirm if the cells knockout for p38α harboring an empty plasmid were also knockout for p38β, and vice versa? Otherwise, all of the experiments are missing this double knockout control, which is essential for the manuscript. If the cells were not double knockout, can you measure expression level of endogenous p38β in p38α knockout cells, and vice versa. Expression of mouse p38α in p38β conditions (4,5,6) would complicate the interpretation.
Response: We agree that the proposal raised by the Reviewer to study double knockout cells (missing both p38a and p38b) is interesting. However, we think that firstly, the goal of the current study was to try and identify p38a-specific and p38b-specific targets. Using p38a and p38b double knock cells would address a different question, namely, what are the targets of p38a+p38b in the cells. Second, unfortunately, the interesting proposal of the reviewer is currently not practical for us, as we do not possess double knockout cells, and even if such cells are obtained the SILAC experiment + analysis would be a year-long. In fact, reviewing the literature we do not think that MEF cells knockout to both p38a and p38b are currently available at all. Unfortunately, we were able to identify, and therefore to quantify, only p38β in p38β variants-transfected p38β(-/-) cells, but not in p38α variants-transfected p38α(-/-) cells, in the proteomics analyses. Furthermore, we were unable to identify and quantify in the proteomics analyses p38α in all the different cell lines. This is in contrast to the western blot analyses in which both isoforms could be observed. This is a common situation for proteomics studies, in which low-abundance proteins remain undetected by LC-MSMS analysis.
- Expression levels of constitutively active variants of p38α and p38β should be similar between the cell lines to make accurate and meaningful comparisons. Based on Figure 1C, it appears that protein levels of p38β are different from p38α. Differential levels of active variants can lead to differences in phosphoproteome.
Response: The Reviewer is correct in pointing out this problem. To achieve as close as possible equal expression levels of the individual MEFs, we sorted the cells according to the levels of the stably expressed p38s using co-expression of GFP from the same vector as an indicator. However, it is possible that somewhat lower levels of expression of p38a are caused by other factors, such as different stability of the proteins (Fig1c). As already mentioned above, these are the limitations of such biological experiments. Yet, we agree with the reviewer that this point should be noted, and we added a note to explain this issue in the Results (lines 191-2) and Discussion (lines 404-409).
- There are substantially higher number of phosphosites that are differentially regulated for p38α active variant compared to p38β active variant. Were the protein expression levels different between these variants?
Response: The protein levels of p38a were in fact somewhat lower than p38b. On the other hand, its level of activity seems to be higher and induced more changes in the phosphoproteome. An explanation about it is added in lines 191-2
- The phosphoproteomics data from anisomycin treatment experiment (Figure S1c) is in stark contrast to that from constitutively active variants of p38α and p38β (Figure S1a-b). As shown in Figure S7, temporal profile of majority of sites is similar if not identical between wild type MEFs, p38α (-/-) and p38β (-/-) suggesting minimal effects and differences between p38α and p38β to anisomycin treatment. As discussed in line 211, large complementarity exists among the p38 variants. This further raises the need for double knockout control or all p38 variant knockout control to truly study the specificity differences between the p38 variants as background expression of other variants complicates the interpretation.
Response: The reviewer is correct in the observation that the partial effect of the anisomycin on the individual p38s’ substrate phosphorylations is relatively smaller than the effect of the intrinsically active p38 variants. We expected this since anisomycin has a strong effect on other stress pathways, in addition to p38, and therefore we looked for the few specific effects of p38 substrates and downstream sites that were distinguishable by the inclusion of individual knockouts from the vast majority of non-p38 substrates that were affected similarly, both in the presence of the individual p38s and in their absence.
- Sites found in Cluster #1045 are interesting since they are differentially regulated in p38β (-/-) suggesting that these sites may be regulated by p38β. How many of these sites are also differentially regulated in the constitutively active variant experiment? Does the profile match between the two datasets for these sites?
Response: This is an excellent suggestion, and we added a sub-figure in Supplementary Figure S7 with a list of proteins that are modified differently in both the intrinsically active mutants and the anisomycin experiments and added an explanation about it in the text, in Lines 236-8. In addition, we added in Supplementary Table S1 columns called ‘cluster’ next to the column ‘significantly changed phosphorylation sites due to anisomycin treatment’ for easier match between the two datasets for all sites.
- These data also raises a concern that the constitutively active variants may have different specificity for substrates compared to wild-type proteins?
Response: Indeed, we were hoping to detect the existing differential specificities of the constitutively active variants, which may point to the specific activities that are caused by the continuous (chronic) activation of these enzymes relative to the wild-type enzymes that are inactive until activation by such reagents as anisomycin. We hope that we succeed in demonstrating a few of these sites. Nevertheless, we fully agree with the reviewer that the activating mutation in the active p38a and p38b might affect not only catalysis but also substrates specificity. Only proteomic studies like this one, that identify potential targets, and analyze the targets one-by-one will tell which are bona fide targets of p38a/b, and which may be specific for the mutants. We wish to add, however, that we performed several studies to address such a potential problem and never found even one case of non-specific phosphorylation by the active variants.
- Please mention how upregulated, downregulated and differentially expressed phosphosites were determined in Figure 2 i.e. fold change and/or p-value cutoffs. It’s very confusing which cutoff are used where.
Response: All the sites listed as ‘changed’ were observed at least 1.5 folds change up or down relative to the wild-type variant or to time point 0 min in the anisomycin treatment as indicated in figure 2 legend. This change is now indicated in Lines 177-181.
- It’s unclear what’s shown in Figure 2D. The figure is not discussed in the results section either.
Response: We apologize for this omission and thank the reviewer for pointing it out. We added an explanation in lines 191-194.
- Section 2.4: p38a and p38b differently affect HSP27 phosphorylation and expression levels: this conclusion may only be partially correct. While constitutively active variant of p38a leads to higher phosphorylation of HSP27 and that HSP27 has low protein levels in p38a (-/-) cells, the temporal profile in anisomycin treatment data (Figure 5b) shows that large complementarity exists. Even though p38a or p38b are absent, HSP27 has similar phosphorylation and protein level profile, essentially contradicting the main statement.
Response: Indeed, some minor difference was observed between the level of HSP27 following the expression of the intrinsically active p38 while the knockout of the p38s had a lower effect on the levels or ratios of this protein (Figure 5). We use this example to emphasize also the differences in the effect of the chronic relative to the transient activation of the studied p38s.
- Section 2.6: Again here with p53: although constitutively active variant shows differential phosphorylation, in vitro kinase assay suggest that both p38a and p38b can phosphorylate p53 at similar levels. How does the profile look in anisomycin treatment data for p53? These data suggest that there may not be differential phosphorylation of p53 by p38a or p38b.
Response: The Reviewer is correct in observing that both p38 phosphorylate p53 (Figure 7) yet, we would like to stress the point that the expression of the intrinsically active p38 variants affects these sites differentially, which is an interesting point, and we believe valuable for the readers. We added panel D to Figure 7 to support this statement and observation and text in Lines 341-346, 366-367.
Minor comments:
- Figure 2B: Spelling error for y-axis label – should be phosphorylation site count
Corrected
- Figure 2C: Spelling error for title – should be signaling
Corrected
- It is challenging to visualize data in Figure 4 left panels because of overlapping labels with the bar plots. Make sure to place the labels appropriately.
Corrected
- Figure 5a is missing y-axis label.
Corrected
- Figures are not discussed chronologically. For example, Figure 5 is discussed in line 187 before Figure 4.
Corrected
- Please provide color bar for Figure S7. Also what does the heatmap represent – fold change, log2 intensity?
Corrected
Round 2
Reviewer 2 Report
Authors have addressed the comments and concerns in the manuscript.
Author Response
Dear Editor
We upload here the updated figures in a zip folder, and the new version of the figures, inserted into the Word file. I will try to upload the figures TIFF file separately if your website will allow that.
We hope that this will solve the issue raised by the Reviewer.
Best Regards
Arie
